# An Empirical Study on the Determinants of an Investor's Decision in Unit Trust Investment

**Sanmugam Annamalah** [1,*], **Murali Raman** [2], **Govindan Marthandan** [2] and **Aravindan Kalisri Logeswaran** [2]

[1] Faculty of Business, SEGi University College, Kuala Lumpur, Wilayah Persekutuan 50100, Malaysia
[2] Faculty of Management, Multimedia University, Cyberjaya 63100, Malaysia
* Correspondence: sanmugam_1@yahoo.com

**Abstract:** Unit trust is a convenient way of investing and a sensible way to build one's wealth in the medium term and subsequently in the long-term. Investment specialists will manage the investments and spread the risks through careful diversification. The basic nature of the unit trust is that it carries a low-level of risks and accordingly determines a lower level of returns compared to other financial instruments. There is a lack of research that empirically investigates the factors that influence an investor's decision in unit trust investment, particularly in a Malaysian setting. The purpose of this study is to analyse the factors that influence an investor's investment decision in purchasing a unit trust. This paper aims to narrow this research gap, whereby financial status, risk taking behaviour, investment revenue and related information are hypothesized to exert statistically significant influences on the investor's decision in unit trust investment. The empirical study uses a quantitative research approach whereby survey data have been sampled from 202 participants using a convenient sampling technique. This research is cross-sectional and uses primary data for analysis. Data analysis has been carried out using multiple regression analysis. The empirical research finds that financial status, risk taking behaviour, and sources of information significantly influence the investors' investment behaviours in unit trusts. However, there was not enough evidence to support the claims that investment return and revenue have a statistical relationship to the investors investment behaviours regarding unit trusts. The findings from this research will have huge implications for investors and for financial institutions. This paper helps fund managers and brokers to understand the behaviours of an individual investor in response to a unit trust. On the other hand, this helps them to better target their customers, and persuade customers to make their investments in a unit trust effectively and efficiently, thereby helping them to manage their financial wealth with less risk but better future prospects.

**Keywords:** mutual fund; unit trust; financial instrument; risk behaviour; investment; financial status; investment revenue; sources of information

**JEL Classification:** K25; O16

## 1. Introduction

### 1.1. Background of Research

Rapid economic growth has reliably increased the income and purchasing power of individuals around the world. Indeed, this growth has increased their desire and need for a wide variety of financial products and services (Van den Burg et al. 2017). In today's fast changing environment, financial products have gained significant attention from individuals, and attracted individuals to make

investment by way of different instruments in order to gain extra income and earnings. In addition, investments have been used as an instrument by individuals as part of their personal financial planning.

Among the different types of investment instruments, the stock market has attracted the most investors, as it provides higher returns to investors. However, it entails a high level of risk to investors. This has resulted in most investors suffering significant losses in the stock market, and has caused investors to take drastic action, such as committing suicide (Agrrawal et al. 2017). Over the past few years, investors have shifted their attention from high risk and high earnings from the stock market to low risk but stable earning instruments, such as unit trusts. According to (Hans 1999), the unit trust is also known as a mutual fund and it is a form of collective investment constituted under a trust deed (Chang et al. 2012). A unit trust pools investors' money into a single fund, which is managed by the fund manager to invest in different profitable projects. A professional fund manager manages the investment funds, and as such, there is a high possibility that there will be high risks that are associated with the unit trust investments. As such, it provides a low-level of earnings. Research conducted by (Shafee 2018) discovered that a unit trust is preferred by most investors, especially working adults. The unit trust provides low risk, but stable earnings and this category of investment is mostly preferred. In addition, unit trust companies can add this instrument as part of their financial planning, to help individuals to steadily accumulate their financial resources to achieve their financial goals.

Unit trusts over the past few years have undergone a slower growth, as most investors are willing to take a high level of risk in exchange for higher returns. Nevertheless, the data published by Central Bank (BNM 2015) has stated that growth in unit trust investment has increased by 0.47% annually. However, the total amount of investment in unit trusts is way below the investment volume in stock markets. In order to increase the investment volume in unit trusts, it is important to determine factors that affect investors' investment behaviour, and this study has been undertaken to identify the significant factors that influence investors' investment decisions in unit trusts. The findings of this study could provide valuable information to various stakeholders such as investors, government and regulatory bodies, as well as unit trust brokers and fund managers.

### 1.2. Research Problem

The financial market in Malaysia has undergone unbalanced developments as most investors prefer to invest in stock markets such as ordinary shares rather than in unit trust investments (Ng 2018). This has brought slower growth in mutual fund investment compared to stock market investment. In the long run, the unbalanced investment will not be able to drive the overall development of the financial market. Dorfman (2018) discovered that the investment volume in the stock market for ordinary share investment is eleven times higher than the total volume investments in the mutual fund market. This has remained a challenge to the government and mooted mutual fund authorities to transform the investment behaviour of investors. On the other hand, Dorfman (2018) has reiterated that investment in the stock market overall does not provide positive returns to investors, as more than 61% of investors suffer losses from investments in the stock market due to below par performances or economic challenges posed by economic volatility and this has resulted in investments in the stock market being unable to offer the returns expected by investors. It is possible to manage income more effectively through personal financial planning for individuals through unit trust investments by reviewing an investor's current financial circumstances, anticipated changes, future goals, and results through a customized plan. In addition, financial literacy among students is important in providing information as well as enhancing financial literacy and wellbeing to invest prudently in the future.

Studies provide intriguing evidence that understanding investment behaviours provides various platforms for different users, and past studies have concluded that there are many factors that influence the investors' investment behaviours. A study undertaken by Waweru et al. (2008) concluded that financial status influences the behaviours of investors in making investments as they treat investment as a form of financial planning to make them wealthier. Therefore, the financial status of the individual determines whether an investor invests their money, in addition to affecting the amounts that they

are prepared to invest in the shares. Tan et al. (2008) findings showed that the financial status of individuals provides key influence as to the financial status of the investor that affects their investment volumes, as well as the level of risks they are willing to take. Investors from low financial status tend to bear little risk, and a unit trust is their preferred investment instrument compared to investors with strong financial status where they are willing to take risk by investing in risky instruments that generate higher returns.

The (Thaler 2015) study indicated that there is a relationship between the risk behaviour of investors and their investment decisions. An (Obamuyi 2013) study indicated that socio-economic factors influence the investment decisions of investors in the capital market. Therefore, such relationships bring changes for different individuals, such as individuals from higher socioeconomic classes or financial backgrounds, who may be able to tolerate higher risks due to their better financial resources compared to investors from low socioeconomic classes who have limited financial resources for investments. Moreover, the risks also determine which investment option they should invest in to commensurate earning with the risk that is being undertaken. In the event of a decline in shares, the losses are within the affordability of the investor. (Leon and Aprilia 2018) observed that there is a relationship between risk taking behaviour and the investment behaviour of investors, such as personality traits, level of education received, income and financial status, as well as preference for taking risks.

According to (Lan et al. 2017), the factors that affect investment behaviours are composite situations, investment techniques, and more importantly the information that investors need for investment purposes. Investors who are able to effectively master market information have an advantage over others as investment knowledge will be more likely to profit (Abul 2019). The research objectives are to identify the relationship between financial status and investors' behaviours for mutual fund investment; to identify the relationship between risk taking behaviour and investors' behaviours regarding mutual funds; to identify the relationship between investment revenue and investors' behaviours regarding mutual funds; and to identify the relationship between availability of information and investors' behaviours regarding mutual funds. The findings could summarise the factors that are important to investors' investment behaviour in mutual funds and enable the formation of a framework to guide new investors to invest in mutual funds, or to help them to choose, select, and consider the appropriate mutual fund for their investment in order to gain stable returns and income. It also delivers valuable information to mutual fund brokers and managers, as they could use these key factors to attract investors to make decisions to invest in a mutual fund, and to boost the performance of mutual fund investments. The findings would also help authorities to improve the development of the mutual fund sectors to provide a better investment environment for unit trusts, and help individual investors realize the benefits and merits of investing in mutual funds.

## 2. Literature Review

### 2.1. Investment Behaviour

An (Obamuyi 2013) study indicated that in addition to socio-economic factors, behaviours also influence investment decisions by investors in the capital market. As such, the behaviour of the individual is the motive and reason for one's act. The behaviour was originally studied and investigated in psychological fields to assist psychologists to understand the reason behind an individual's action, and the reasons behind it. Various motivational theories and models were developed to help motivate individual's acts (Loewenstein 2000). Investment behaviour is the process of considering different factors that influence an investment decision for a specific investment instrument. Investment behaviour can be influenced by a wide range of factors and forces, and these factors can be divided into personal characteristics, such as personalities, self-motivation, and other traits (Merikas et al. 2003). In addition, the behaviour of an investor can be influenced by external factors and forces such as the general economic environment, a stock's past performance, and other related aspects (Tavakoli et al. 2011).

Understanding investment behaviour can be important to different users, such as for new investors to select suitable stocks for investment. In addition, it also helps the investment broker to effectively and efficiently attract investors' investment decisions by offering an investment instrument that meets the needs and expectations of the investors (Barnea et al. 2010).

A study by Waweru et al. (2008) study concluded that financial status greatly influences the behaviours of investors when making an investment. Therefore, the financial status of the individual generates influence on whether the investor invests their money, and, in addition, effects the amounts that they invest in the shares. Tan et al. (2008) also investigated investment behaviours and found that the financial status of the individuals provides key influence and impacts the investors' behaviours in unit trust investments. They also stated that financial statuses of the investors are likely to affect their investment amounts, as well as determine the level of risks that they are willing to take. Investors from low financial status tend to bear less risk, and a unit trust is their preferred investment instrument compared to investors from better financial status who are willing to take greater risks to invest in risky instruments that generate higher returns.

Various studies have investigated the effects of risk behaviour and investors' investment behaviour and the findings identified that there is a relationship between the risk behaviour of investors and their investment behaviour (De Bondt and Thaler 1987; Thaler 2015). The authors argued that such relationships vary with different individuals, such as individuals from a higher socioeconomic classes or financial background are able to tolerate more risks due to their possessing better financial resources compared to investors from lower socioeconomic classes who have limited financial resources for making investments. In addition, it was also concluded that the term risk was used by investors to determine the investment options that provides higher returns. However, if the stock goes down in price, the losses are within the affordability of the investors. Mak (2017) has also investigated the relationship between risk and investment behaviour of individual investors and observed that a relationship exists between risk taking behaviour and the investment behaviour of investors. It was also found that not only does risk taking behaviour affect investors' investment behaviour, but other factors also facilitate the risk taking behaviour of the investors, such as personality, level of education, income and financial status, as well as preferences for specific shares as well risk taking behaviour.

Research conducted by Khan et al. (2015) discovered that investment revenue or expected return for investors has a relationship with the investment behaviour of the investors. The researchers argued that the investors expected return to help them to filter and select an instrument that fits their requirements. Investors will consider the options of past returns to meet their expected return to make investment decisions. In addition, investors are also attracted by investment options that deliver extraordinary returns to investors. Jagongo and Mutswenje (2014) found that a critical relationship exists between the availability of information on the stock and investors' investment decision. They argued that the investment option offers a wide range of information that will help investors to better analyse the performance of the stocks, and a decision is more likely to be made based on that information. Stocks with less information would have negative effects on investors since the stock contains higher risk, as the market provides inadequate information for the investors to make an informed investment decision.

## 2.2. Financial Status

The financial status of an individual is the most important aspect in influencing their behaviour for the purpose of investments. It is important to understand that the financial status of an individual indicates the amounts of saving that the investor has, as well as their fixed income, such as wages that the investor receives on a monthly basis, and these financial resources are the most fundamental when it comes to supporting one's investments (Aranda-Uson et al. 2019). The better the financial status of an individual investor, the more likely they are to show positive behaviour when investing their money into an investment stock or instrument, and they also tend to invest in large amounts of money (Aranda-Uson et al. 2019). In addition, Herranz González and Martínez-Carrascal (2017)

stated that financial status does influence the investment decision of investors. Most rational investors will use their financial status to determine their behaviours for investment, but with current trends, numbers of investors who invest in larger amounts have increased, although they are in the category of middle-income earners. They are willing to invest, in order to enhance their earnings and incomes. A study undertaken by Waweru et al. (2008) concluded that financial status has an influence on the behaviours of investors regarding their making investments and treating investments as a form of financial planning to make them wealthier. Therefore, the financial status of the individual generates an influence on the investor's desire to invest their money, and it also affects the amounts that they invest in stocks and shares. Tan et al. (2008) have also investigated investment behaviours for unit trust investment and the findings showed that the financial status of individuals provides a key influence and impacts on the investors' behaviours. In addition, they stated that the financial status of the investors will more likely affect their investment amounts, as well as the level of risks that they are willing to take. Investors from low financial status tend to undertake little risk, and a unit trust is their preferred investment instrument compared to investors of better financial status who are willing to take risk to invest in risky instruments that generate higher returns.

**Hypothesis 1 (H1).** *There is a positive relationship between investors' financial status and investment behaviours.*

### 2.3. Risk Taking Behaviour

Risks are basically uncertainties that happen in all aspects of life, and therefore there is a need to identify these risks in order to ensure the successful accomplishment of goals. Risk-taking behaviour is the ability of individuals to take risk for their investment or any other act. A risk-taking attitude or behaviour is important in making investments, and risk-taking behaviour is also used by individuals to make the selection of their stocks (Leon and Aprilia 2018). According to Trang and Tho (2017), the investment instrument carries a different level of risks to investors, such as investment in an ordinary share that generates higher risk in addition to delivering higher returns and earnings. Certainly, a unit trust entails lower risk and at the same time the return is considered low to the investors. Wright (2017) stated that there is a positive relationship between risk and earnings in investment. Understanding risk-taking behaviour is important to investors themselves and investment brokers. Knowing risk-taking attitudes will help the investors themselves to make selection of the investment instrument that are within their affordability and risk range. This helps to prevent over-reaction when investors suffer losses. On the other hand, risk-taking behaviour also helps investment brokers to introduce investment options to customers that are more affordable and to make their desired investment a profitable one.

Thaler (2015) investigated the effects of risk behaviour and investors' investment behaviour and the findings stated that there is a relationship between the risk behaviour of investors and their investment behaviour. The authors argued that such a relationship changes according to the individuals as individuals from higher socioeconomic classes or financial backgrounds are able to tolerate more risks due to their financial resources they possess compared to investors from low socioeconomic classes that have limited financial resources. The authors also conclude that risks are usually used as a scale by investors to determine which investment option they should invest in, and to make earnings, and whether the losses are within the affordability of the investors. Another study proposed by Van Raaij (2016) observed that a relationship existed between risk-taking behaviour and the investment behaviour of investors. Apart from risk-taking behaviour affecting investors' investment behaviour, there are various other factors that facilitate the risk-taking behaviour of the investor, such as personality, level of education received, income and financial status, as well as preferences for the specific stock in taking risks.

**Hypothesis 2 (H2).** *There is a positive relationship between investors' risk-taking behaviour and investment behaviours.*

### 2.4. Investment Revenue

Investment revenue is the possible earnings that investors are estimated to earn from investing in specific investment options (Lusardi and Mitchell 2017). Every investor has his or her expected returns upon their investment, and investors will invest based on their expected earnings, and therefore they will look for suitable stocks that have generated high earnings in the past to match their expectations (Aregbeyen and Mbadiugha 2011). Investment revenue has been used by investors as the key criteria for selecting stocks and instruments for investments. Khan et al. (2015) discovered that an investment's revenue or expected return for investors has a significant relationship with the investor's investment behaviour. This is due to the expected return, as the expected return assists the investors to filter and select the instruments that fit their requirements. Investors will seriously consider the options of the past returns or options that meet their expected return to make an investment decision. Also, investors are attracted by those investment options that deliver extraordinary returns to investors.

**Hypothesis 3 (H3).** *There is a positive relationship between investment revenue and investment behaviours.*

### 2.5. Availability of Information

Information is important for individuals to decide on different aspects, especially for investment decisions, whereby investors make their investment decisions based on the information of the instruments, such as the company's past financial performance, the distributed dividends, and the past market share price movements (Abul 2019). There are many factors affecting the investor's decision-making, and information is an important consideration because it plays an important role in decision-making as it affects investors' consideration and decision making for investing (Sarwar and Afaf 2016). Lubis and Sudarisman (2017) study found that a critical relationship existed between the availability of information om the stock and investors' investment decisions as investment options offered with a wide range of information that will help the investor to better analyse the performance of the stock, in which the decision is more likely to be made. However, stocks with less information provided, negative feelings are generated in investors, feelings which suggest that the stock contains higher risk in investment. This is due to the fact that investors will not be able to exploit the information available to make informed investment decisions (Khan et al. 2017). As for the mutual fund purchase decisions, information includes formal and informal and the information sources are available for investors as a guide for investments. Investor's knowledge about the expenses and risks associated with investment in the mutual funds can make mutual funds options more attractive, useful and helpful to the investor as the investors are able to select mutual funds schemes in a better way in accordance to risk tolerance (Shanmugham 2000).

**Hypothesis 4 (H4).** *There is a positive relationship between availability of information and investment behaviours.*

### 2.6. Underpinning Theories

Two related behavioural theories which are based on the cognitive behaviour model and the planned behaviour model have been used in this study. Planned behaviour theory focusses on the individuals that make logical, reasoned decisions to engage in specific actions by assessing the information that is available (Ajzen 1991). The theory emphasizes the probability of success and better performances, wherein the perceptions of individuals are crucial in making certain investment

decisions (Ajzen 1987). It also shows the strength of the attempt of the individuals to engage in actions and how much control the individual has over the investments, and thereby influences the decision of whether to invest. In this study, the greater the perceived behavioural control; the stronger that person's intention to invest. The greater the perceived behavioural control, the more favourable the person's attitude towards investments, and thereby the stronger the person's desire to invest. In addition, intentions represent a person's motivation in the sense of conscious plan or decision to exert effort to invest (Conner and Sparks 2005).

Cognitive models of behaviour's first process involve 'thought'. When an individual perceives a product or service, he or she has the 'thought' of the product, such as whether the product is able to provide critical functions and possess features that can benefit the customer, or the 'thought' of whether it is worth purchasing the product (Kotler and Keller 2011). The second process involved is the cognitive model of behaviour, which refers to 'feelings'. 'Feelings' are created by the thought. In other words, the thought is the process of information to determine the goodness and badness of the products or services, and the feeling is the consequence of comparing the strengths and the weaknesses (Zeithaml et al. 1996). A positive feeling could arise when the benefits outweigh the limitations, and negative feelings will occur when the limitations or costs are greater than the benefits of the products or services. Most importantly, the feelings will formulate the ultimate perceptions of the individuals for the products or services, such as investors being aware of the nature of unit trust investment, as it will form the perception in the mind of investors that unit trust provides low risk, better earnings, and a stable income. As the perception is formed with the feeling of individuals for a products or services, the feeling will then influence the behaviours of the individuals (Zeithaml 1988). For instance, a customer is more likely to purchase a product that has positive perceptions. Therefore, the higher the positive perceptions for a product, the stronger the behaviour of the individuals, which leads to the strong purchasing intention for the precise products or services. Figure 1 illustrates the conceptual framework of the research which represents the factors that contribute to the development of specific variables described in the literature.

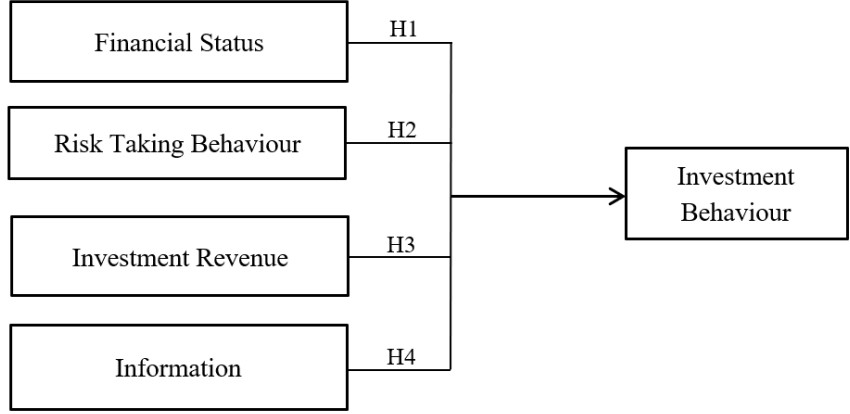

**Figure 1.** Conceptual Framework.

## 3. Research Methodology

The study uses descriptive statistics, a method which is suitable for research that attempts to describe events from one point to another, and this method is mostly used in studies that are attempting to understand a certain phenomenon over a certain period of time. This research investigates factors that influence the individual investors' investment behaviours for mutual fund investments. Factorial information was chosen from past studies to focus this research on investigating the perceptions of the investors' investment behaviour towards mutual fund investment. A quantitative research method is the widely used approach, and it is a method that uses numerical data as the input for the research, using statistical methods to analyse the data (Creswell 2009). This study examines the relationship between independent and dependent variables in order to understand the influences of these factors

on the investors' investment behaviours for unit trusts. Therefore, this research is more suited to a quantitative research method, as it could help the study to accurately examine and express the relationship in numerical form for better understanding. The respondents in this study are individual investors aged between 25 to 60 years. The sample size designed for this study is 250 participants. This research uses the non-probability sampling method, which is a convenient sampling technique allowing the researcher to select and choose participants at the best convenience of the researcher. For convenience sampling, a sample size between 200 to 500 respondents is preferred as it would indicate more a reliable sample and prove the validity of the results (Churchill 1991). The convenience sampling method was used in this study because of the advantages of it being the least expensive and least time-consuming (Malhotra 2004; Park and Sullivan 2009; Sekaran and Bougie 2016). This helps us to reduce the time spent on the selection of participants. Primary data collection is used in this study, and hence the data collection instrument used in this study is a survey questionnaire. The survey questions were adopted from past research studies and consists of close-ended questions which allow the researcher to convert the data into numerical values to apply statistical methods to analyse the data (Dodge 1985). The survey questionnaire contains two sections. The first section provides the demographic profile which examines the personal backgrounds of participants, such as gender, age, income, occupation, and others. The second section of the survey questionnaire examines the agreement level of respondents toward some aspects as well as the investment behaviour for unit trust investments. A copy of the final questionnaire is appended as Appendix A. The measurement scale used in this research study is the five Likert scale, and the reason for the use of five Likert scale is that it copes with the adoption of close-ended survey questions, as well as assisting researcher to later convert options into numerical values with SPSS software.

### 3.1. Data analysis and Findings

There were a total of 250 participants chosen for this research study, and there were sufficient amounts of survey questionnaires distributed to these 250 participants. The survey resulted in a total of 202 responses, indicating a total response rate of 80.8%. The reliability test was performed by using the Cronbach's Alpha test, and this was to examine the internal consistency among the data collected in this research. In order to determine whether the data are reliable, the results produced from the Cronbach's Alpha needed to be above 0.7 to show that the data is reliable (Sekaran and Bougie 2016). There were five variables included in this research, where four are independent variables while one is a dependent variable. Each variable contains an equal number of five items. The Cronbach's Alpha obtained for variables in terms of financial status, risk taking behaviour, investment revenue, source of investment information, and investment behaviour are 0.709, 0.717, 0.742, 0.773, and 0.739, respectively. Table 1 provides the results that the measurement variables that are above and higher than 0.7, showing that the data collected in this research are reliable in nature to ensure the reliability of the research findings.

**Table 1.** Reliability test of data.

| Variable | No of Item | Cronbach's Alpha |
| --- | --- | --- |
| Financial Status | 5 | 0.709 |
| Risk Behaviour | 5 | 0.717 |
| Investment Revenue | 5 | 0.742 |
| Source of Investment Information | 5 | 0.773 |
| Investment Decision | 5 | 0.739 |

The validity test is to determine whether the instrument obtained the correct answers for the questions. The result of KMO and Barlett's should be higher or above 0.6 to show that the data collected are valid for the study (Pallant 2013). Table 2, presented below, shows the validity test of KMO and Barlett's for the data collected in this study. The KMO and Barlett's results obtained for financial status, risk taking behaviour, investment revenue, source of investment information, and investment

behaviour are 0.687, 0.792, 0.771, 0.707, and 0.642, respectively, and all are significant as shown in Table 2. The results obtained are higher than the required 0.6 to show that data collected in this research are valid.

**Table 2.** Validity test of data.

|  | **KMO and Barlett's** | **Sig** |
| --- | --- | --- |
| Financial Status | 0.687 | 0.000 |
| Risk Behaviour | 0.792 | 0.000 |
| Investment Revenue | 0.771 | 0.000 |
| Source of Investment Information | 0.707 | 0.000 |
| Investment Decision | 0.642 | 0.000 |

A normality test was conducted to determine whether the data collected are normally distributed or non-normally distributed. Data that are normally distributed indicated that there is consistency of the data collected, and in other words there are minimal amounts of mistakes or errors contained in the data. The Q-Q plot pairs up corresponding quantiles from the samples, and in this case, it is between independent and dependent variables that determine whether the sample data collected are normally distributed by inspecting the scatterplot. If they are normally distributed, the points in the scatterplot should lie close to the line (Doyle 2010). In order to examine the normality of the data for this study, the graphic method of a normal Q-Q plot was used, and the results are presented in Figure 2.

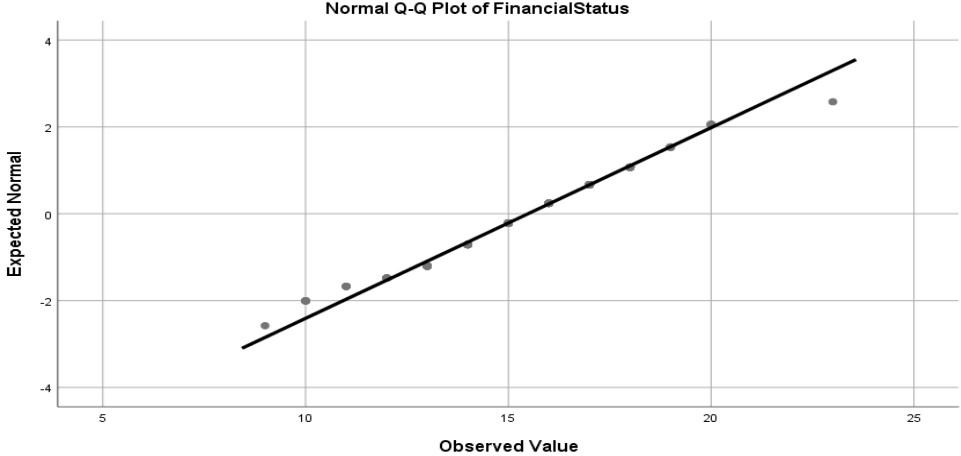

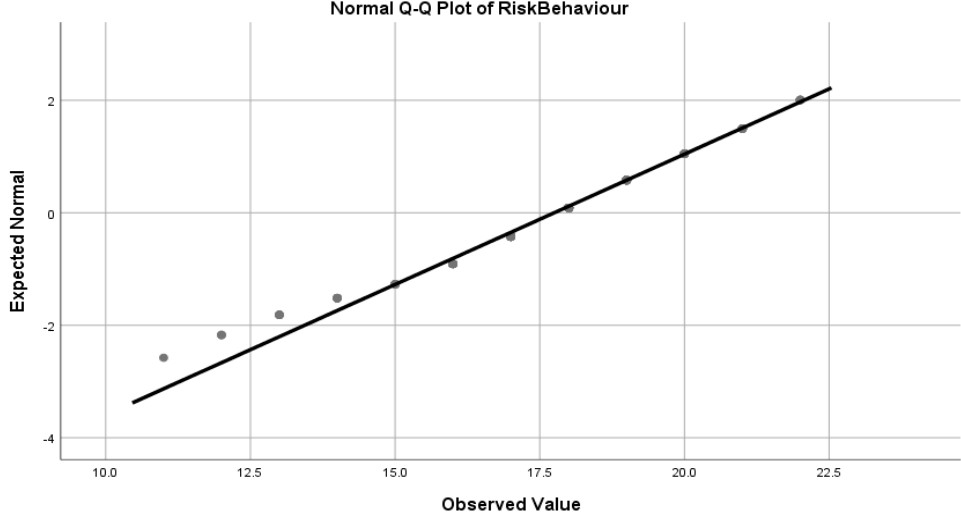

**Figure 2.** *Cont.*

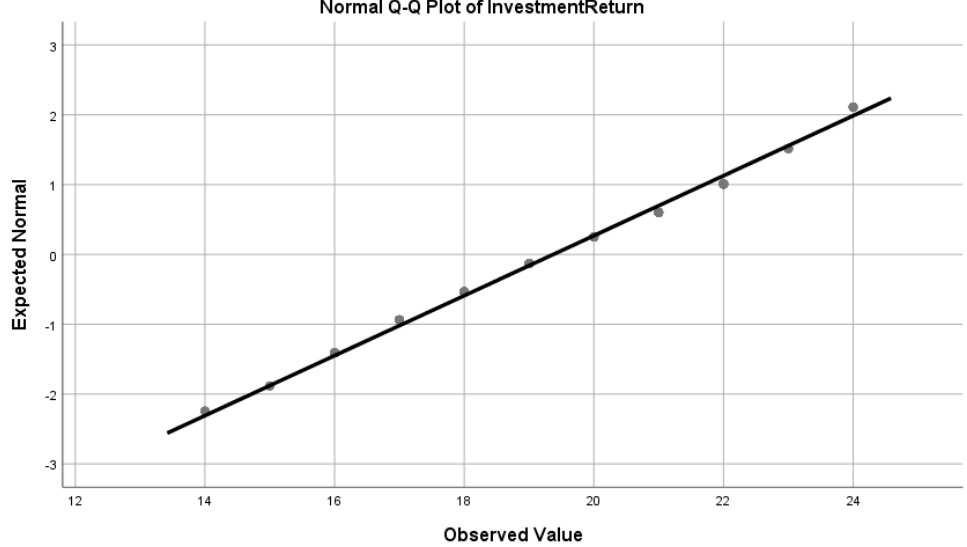

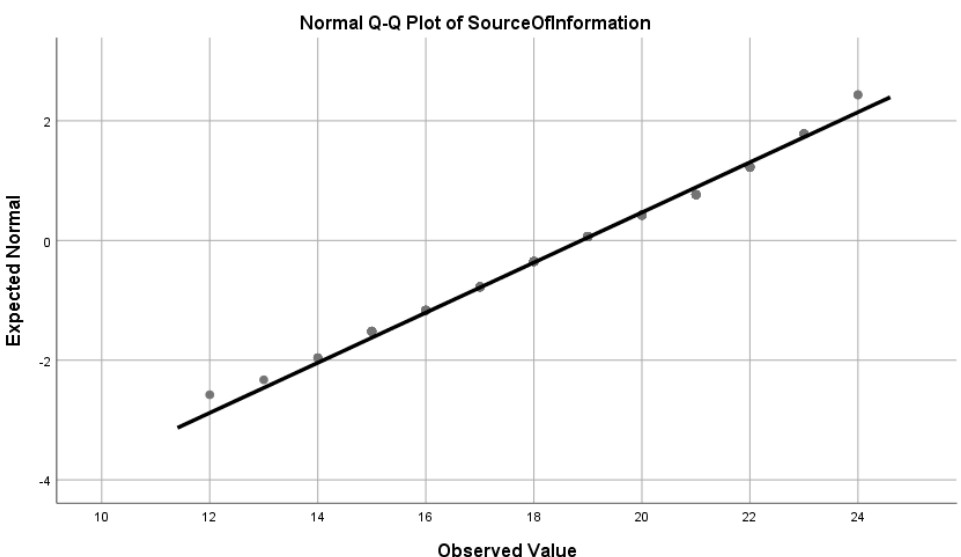

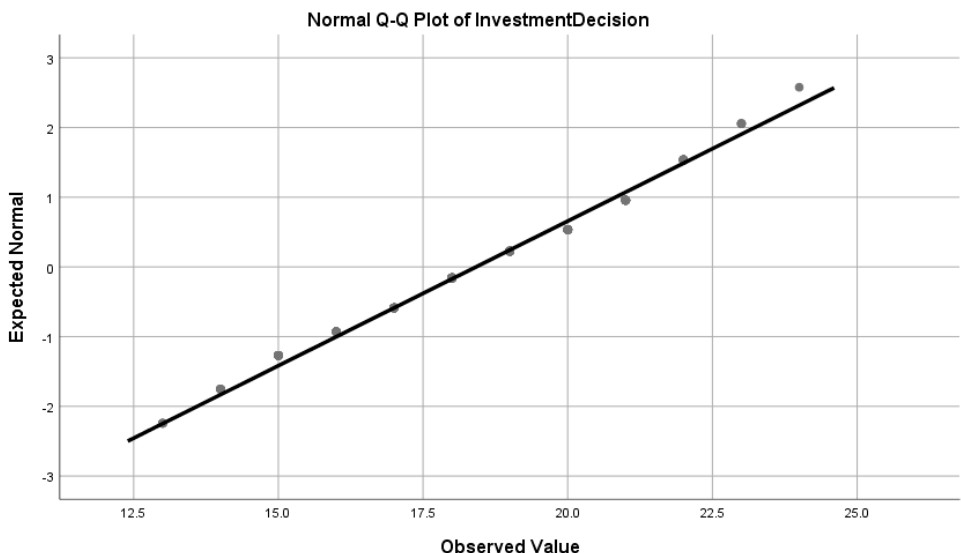

**Figure 2.** Theoretical Quantiles of the Normal Distribution.

Based on the observation of the above normal Q-Q plots, it is observed that data collected under each variable in this study are normally distributed. This was shown on plots in all graphs that they are mostly on the centred line, and plots are closely distributed to the line to show they are normally distributed rather than abnormal distributions. Hence, the data collected in this research are normally distributed to ensure its data quality.

The demographic profile of participants is essentially important for this study. According to Creswell (2009), the demographic profile of participants could help a researcher to determine the behaviours and thoughts of participants, and with the linking of the related demographic profile, it helps to explain the behaviours of the participants in their investment behaviour. For instance, income level will determine the risk-taking behaviour of participants. Gender is one of the important aspects to be investigated. Many psychologists have concluded that there are biological differences between male and female in their behaviours, as well as in decision-making processes (Ngun et al. 2011). The female individuals tend to be picky and they undergo a long decision-making process to make their decisions as they consider various factors to make decisions. Whereas, male individuals tend to be easy going, and spend less time on making decisions, at the same time there are also other factors taken into consideration in arriving at a decision.

Table 3 shows the percentage of male and female participants involved in this study. Male participants in this research account for 43.1%, while female participants are at 56.9%. This was done after studies indicated that females are better at investments compared to males (Cannivet 2018; Collinson 2018). Despite, the fact that there are more female participants, the findings of this research could fairly represent the investment behaviours of both gender groups of investors. Age is another important personal feature that needs to be investigated in this study. Age has the tendency to change individuals' thought and thinking and tends to show different behaviours. The reason for the change in behaviour of individuals at different age levels can be attributed to their knowledge and experiences acquired, and this will eventually lead to variation in behaviour and thinking patterns. Table 4 shows the distribution of respondents according to their respective age groups.

**Table 3.** Respondents gender status.

|  | Gender | Frequency | Percent |
|---|---|---|---|
| Valid | Male | 87 | 43.1 |
|  | Female | 115 | 56.9 |
| Total |  | 202 | 100.0 |

**Table 4.** Respondents' Age Groups.

|  | Age | Frequency | Percent |
|---|---|---|---|
| Valid | 21–30 | 17 | 8.4 |
|  | 31–40 | 76 | 37.6 |
|  | 41–50 | 92 | 45.6 |
|  | 51–60 | 15 | 7.4 |
|  | 61 and above | 2 | 1.0 |
| Total |  | 202 | 100.0 |

Table 4 displays the percentage of respondent's behaviour and thinking pattern of the different age groups. Based on the data, most of the participants are predominantly in the age group of between 41 to 50 years, as they reflect 45.6% of the total participants in this study. This is followed by 37.6% of participants who are between the age of 31 to 40 years; 8.4% of participants are between 21 to 30 years old, 7.4% of participants are between 51 to 60 years old, and only 5% of the participants are above 61 years. This clearly indicated that the findings are representative of investors from all age groups. The status of the individual is also an important aspect in this study, as individuals who are single are

less willing to make investment, as they only have sufficient funds for self-spending, and they do not have the affordability to make long-term financial planning. Individuals who are married have strong desires for investment, as they need to make more money with the limited funding in the future to support their realisation of their dreams and raise their children within their living expenses.

For the perspective of the employment status of participants in Table 5, the majority of participants are employed, and they account for 57.3%. In addition, 20.3% of participants are self-employed which indicates that they are involved in businesses. Four and a half percent of participants are unemployed, while 14.4% of participants are students. Respondents who have retired and housewives are at 3.5%. Therefore, the findings of this study are also representative of participants at different employment status. Educational level refers to the knowledge that the individuals possess at certain level that influences their decisions. Respondents with a higher level of educational backgrounds are able to think logically and able to make informed decisions.

**Table 5.** Respondent's employment status.

| Status | Frequency | Percent |
|---|---|---|
| employed | 116 | 57.3 |
| self-employed | 41 | 20.3 |
| unemployed | 9 | 4.5 |
| student | 29 | 14.4 |
| retired/housewife | 7 | 3.5 |
| Total | 202 | 100.0 |

The results presented in Table 6 show the distribution of respondents' educational background. Based on the results, 29.7% of participants have graduated from high schools, referring to their highest qualification possessed. In addition, a vast majority of participants at 42.6% have bachelor degrees as their highest qualification. Eight point four percent of participants obtained their masters, while 3.0% of participants received their PhD. Finally, 16.3% of participants have other qualifications, such as professional certificates. Hence, participants involved in this research are well-educated, and this also ensures that the responses provided by participants are reliable.

**Table 6.** Respondents' educational level.

| Education | Frequency | Percent |
|---|---|---|
| High school and below | 60 | 29.7 |
| Undergraduate | 86 | 42.6 |
| Masters | 17 | 8.4 |
| Phd | 6 | 3.0 |
| Others | 33 | 16.3 |
| Total | 202 | 100.0 |

Income level is the most important aspect in this study, as it reveals the financial status of the individual, and the ability to absorb risks in their investments. Individuals with a high income are able to invest large amounts in investment instruments compared to low income groups, whom are only able to afford low risks. Table 7 shows the income distribution of participants involved in this research. Thirty-two-point-two percent of the respondents are in the category of earning monthly income of RM8001 to RM10,000. This is followed by respondents earning RM3000 and below comprising 26.7%. 23.8% of the participants receive a monthly income of between RM5001 to RM8000, and only 8.9% of participants are able to make a monthly income of above RM10,001. Nevertheless, all participants in this study are well salaried to support their investment activities.

**Table 7.** Respondents' income levels.

| Income | Frequency | Percent |
|---|---|---|
| 3000 and below | 54 | 26.7 |
| 3001–5000 | 17 | 8.4 |
| 5001–8000 | 48 | 23.8 |
| 8001–10,000 | 65 | 32.2 |
| 10,001 and above | 18 | 8.9 |
| Total | 202 | 100.0 |

### 3.2. Financial Status

The financial status of the individual is the most important aspect in influencing one's behaviour to invest. It is important to understand that the financial status of the individual indicates the amount of savings that investors have, as well as their fixed income, such as wages that investors have on a monthly basis, as these financial resources are the most fundamental in supporting their investments (Tsaurai 2015). Individuals with good financial standing are more likely to show positive behaviours to invest their money in stock investments, and they also tend to invest in large amounts of money. Table 8 shows the financial status and the mean range for items ranged between 2.32 and 3.60. This shows that the overall mean score for this variable is at the range of positive agreement and indicates that financial status is important in determining their investment behaviour.

**Table 8.** Descriptive statistic of financial status.

| Descriptive Statistics | N | Mean | Std. Deviation |
|---|---|---|---|
| Investors rate of investment are based on current financial status and incomes. | 202 | 3.00 | 1.111 |
| Investors do not invest in instruments that exceed their current financial status. | 202 | 2.32 | 1.060 |
| Investors make investment that is within their current affordability. | 202 | 3.37 | 1.144 |
| Investors have monthly portion contributed from incomes for the purpose of investments. | 202 | 3.60 | 1.008 |
| The investment amounts varies according to investors income. | 202 | 3.18 | 1.162 |
| Valid N (listwise) | 202 | | |

### 3.3. Risk Taking Behaviour

The risk-taking behaviour refers to the ability of the individual towards assuming risks for their investments or any of other acts that is related to investments. Risk-taking attitude or behaviour is important in making investment, and therefore, risk-taking behaviour is also used when individuals make their selection of stocks. Table 9 shows the risk taking behaviour mean scores for its items, and the mean scores range from 2.80 to 4.33, and most of items' mean scores fall within the range of between 2.80 and 4.33, and this reveals that risk taking is an important aspect for the investment behaviour to purchase unit trust.

**Table 9.** Descriptive statistic of risk-taking behaviour.

| Descriptive Statistics | N | Mean | Std. Deviation |
|---|---|---|---|
| Investors consider the risk of each type of instrument to make decisions for investments. | 202 | 4.33 | 0.701 |
| Investors invest in unit trust because it carries a lower level of risk compared to other similar investments. | 202 | 3.23 | 0.907 |
| The level of risk determines the return from the investment. | 202 | 2.80 | 0.999 |
| Investors took personal risk assessment test to suits the risk-taking abilities. | 202 | 3.40 | 0.968 |
| The risk of the instrument determines investors investment decisions. | 202 | 4.00 | 0.726 |
| Valid N (listwise) | 202 | | |

*3.4. Investment Revenue*

Investment revenue is the possible earnings that the investors are expected estimated to earn from investing in the specific investment options. Each investor will have his or her expected returns upon their investment and based on their expected earnings will look for suitable stocks that have generated past earnings matching their expectations. The investment revenue is used by investors as the key criteria for selecting stocks and instruments for investment. Table 10 shows the mean scores and values for investment revenue variable, and the mean scores range from the lowest of 3.54 to 4.31 and this shows that most of respondents agreed and strongly agreed to the items that are included in this variable.

**Table 10.** Descriptive statistic of investment revenue.

| Descriptive Statistics | N | Mean | Std. Deviation |
|---|---|---|---|
| Investors have their own expected rate of return for the investment. | 202 | 3.58 | 1.248 |
| Investor's use their expected rate of return as the benchmark for choosing investment options. | 202 | 3.54 | 0.967 |
| Investors will invest in an investment that gives highest return. | 202 | 4.21 | 0.702 |
| Unit trust has the nature to deliver lower investment revenue to investors. | 202 | 4.31 | 0.912 |
| The investment revenue affects investor's investment decision | 202 | 3.72 | 1.147 |
| Valid N (listwise) | 202 | | |

*3.5. Availability of Information*

Information is important in decision making for individuals for different aspects, especially on investment decisions. Investors make their investment decisions based on the information of the instruments, such as the company's past financial performance, the dividends distributed, and the

market share price movement over the past months and years. Indeed, the availability of information affects investors' consideration and decision making for investing in any form of investments. Table 11 shows the mean scores for items in the availability of information. The mean score ranged from 2.99 to 4.21, and this shows that information is important to investors in making decision to invest.

**Table 11.** Descriptive statistic of availability of information.

| Descriptive Statistics | N | Mean | Std. Deviation |
|---|---|---|---|
| Investors prefer to use information that is published by well-known organizations such as investment banks. | 202 | 3.87 | 1.009 |
| Investors use information that are published and analysed by others to assist in investment decision making. | 202 | 4.21 | 0.914 |
| The past revenue and prices of the instrument is investors prior focus for determining the potentials of the instrument. | 202 | 2.99 | 1.172 |
| The availability of information for a specific instrument affects investors instrument selection | 202 | 3.87 | 0.994 |
| The source or channel of information for the instrument affects investor's investment decisions. | 202 | 3.92 | 1.050 |
| Valid N (listwise) | 202 | | |

### 3.6. Investment Behaviour

Investment behaviour is the process of consideration based on different factors to make the right investment decisions for specific investments. Investment behaviour can be influenced by a wide range of factors and forces, and these factors can be divided into personal characteristics, such as personality, self-motivation, and others. Thus, the behaviour of the investor can also be influenced by external factors and forces such as the general economic environment, stocks past performances, and other related factors. Table 12 presents the mean scores for items included in the investment behaviour. The mean scores ranged from 3.44 to 3.88 and the mean scores of all the items shows that investors have the positive investment behaviour for unit trust.

Table 13 shows the results of Pearson correlation analysis. It shows that all independent variables have a positive but low level of correlation with the investment behaviours for unit trust. Table 14 shows that all the factors have R values lesser than 0.2 to barely show the significant positive correlation. Multiple regression analysis is also employed in this research. Multiple regression analysis in this research helps to assess whether there exists a statistical relationship between each of these factors and the investors' investment behaviours for mutual fund. There are two results to be looked at in the multiple regression analysis. First, the R square value shows that the total effects given by independent variables in this research on the dependent variable, and it represents the percentage of change that dependent variable that will be affected. On the other hand, the significance value helps to assess the relationship between each independent variable and dependent variable, to gauge whether there is the statistical relationship in between. When the significance value is less than 0.05, there is the statistical relationship in between or vice versa. Most importantly, the significance value is also used to determine whether the research hypothesis will be accepted or rejected. Risk behaviour has the strongest relationship among all, followed by the source of information, financial status and investment

return. Pearson correlation analysis shows that there is a positive relationship between the variables of the study and investment decisions. According to Table 13 results, financial status and sources of information were found to be significantly related to investment decision. Therefore hypothesis 1 and 4 are accepted in this study.

**Table 12.** Descriptive statistic of investment behaviour.

| Descriptive Statistics | N | Mean | Std. Deviation |
|---|---|---|---|
| Investors will continue to invest in unit trust in the future. | 202 | 3.71 | 1.123 |
| Investors will introduce friends and family members to invest in unit trust. | 202 | 3.88 | 1.005 |
| Investors invest in unit trust because it gives stable returns and revenues. | 202 | 3.44 | 1.097 |
| Investors will bear lower risks but earn higher returns by investing in unit trust rather than depositing money in banks for low interests. | 202 | 3.52 | 1.194 |
| The unit trust is the part of investor's long term personal financial planning. | 200 | 3.78 | 0.987 |
| Valid N (listwise) | 200 | | |

**Table 13.** Pearson correlation analysis.

| | | Financial Status | Risk Behaviour | Investment Return | Source of Info | Invest Decision |
|---|---|---|---|---|---|---|
| Financial Status | Pearson Correlation | 1 | −0.035 | 0.098 | 0.074 | 0.046 |
| | Sig. (2-tailed) | | 0.001 | 0.001 | 0.001 | 0.001 |
| | N | 202 | 202 | 202 | 202 | 200 |
| Risk Behaviour | Pearson Correlation | −0.035 | 1 | 0.042 | 0.215 ** | 0.088 |
| | Sig. (2-tailed) | 0.001 | | 0.001 | 0.001 | 0.216 |
| | N | 202 | 202 | 202 | 202 | 200 |
| Investment Return | Pearson Correlation | −0.098 | 0.042 | 1 | 0.026 | 0.100 |
| | Sig. (2-tailed) | 0.001 | 0.001 | | 0.001 | 0.001 |
| | N | 202 | 202 | 202 | 202 | 200 |
| Source of Info | Pearson Correlation | 0.074 | 0.215 ** | 0.026 | 1 | 0.160 [1] |
| | Sig. (2-tailed) | 0.001 | 0.001 | 0.001 | | 0.001 |
| | N | 202 | 202 | 202 | 202 | 200 |
| Invest Decision | Pearson Correlation | 0.046 | 0.088 | 0.100 | 0.160 [1] | 1 |
| | Sig. (2-tailed) | 0.001 | 0.001 | 0.001 | 0.001 | |
| | N | 202 | 202 | 202 | 202 | 200 |

** Correlation is significant at the 0.01 level (2-tailed). [1] There is a significant relationship found between the investment decision and source of information as confident investors increase their trading using specialized source of information.

**Table 14.** Model Summary.

| Model | R | R Square | Adjusted R Square | Std. Error of the Estimate |
|---|---|---|---|---|
| 2 | 0.210 [a] | 0.144 | 0.024 | 2.37901 |

[a] Predictors (Constant), Source of Information, Investment Return, Financial Status, Risk Behaviour.

The R square value shows that the total effects from independent variables of financial status, risk-taking behaviour, source of information, and investment revenue or return on the investment behaviour at 14.4%. This shows the low level of relationship between these factors to the investment behaviours. Looking at the significance value, this shows that investment revenue or return found to have no statistical relationship with the investment behaviour, as the significance value is 0.110 that is higher than 0.05. From Table 15, it shows that financial status, risk behaviour and source of investment information have significant values of less than 0.05 which explains that they have a statistical relationship with the investment behaviour for unit trusts. However, investment revenue is statistically insignificant based on the results, and therefore, hypothesis three is rejected, while hypothesis one, two, and four are accepted.

**Table 15.** Coefficients.

| Model | | Unstandardized Coefficients | | Standardized Coefficients | t | Sig |
|---|---|---|---|---|---|---|
| | | B | Std. Error | Beta | | |
| 1 | (Constant) | 17.698 | 2.536 | | 6.978 | 0.000 |
| | Financial Status | 0.088 [a] | 0.075 | 0.074 | 0.098 | 0.049 |
| | Risk Behaviour | 0.064 | 0.080 | 0.057 | 0.791 | 0.030 |
| | Investment Revenue | −0.117 | 0.073 | −0.113 | −1.605 | 0.110 |
| | Source of Investment Information | 0.156 | 0.073 | 0.155 | 2.148 | 0.033 |

[a] Dependent Variable: Investment Decision.

## 4. Discussion of Findings

Four factors were examined in this research to determine their influence and the relationship on the investors' investment behaviours for unit trusts. The data were collected from 202 participants, and the findings of the study discovered that the investment return and revenue have no statistical relationship to the investors' investment behaviours. Financial status, risk-taking behaviour, and sources of information were found to have significant relationship of influence on the investors' decision in unit trust investments. Among these factors, the availability of information has a strong relationship on the investors' behaviours, followed by the risk-taking behaviour, and financial status of individuals. Nevertheless, the overall significance influencing factors found in this study are found to be relatively low when it comes to the transformation of investment behaviours to purchase unit trust products.

The findings of this study were consistent with previous studies. This study has discovered that the financial status of investors has a relationship that influences the investors' investment behaviours for unit trust investment. Herranz González and Martínez-Carrascal (2017) stated that rational investors will follow the traditional pattern to use their financial status to determine their behaviours for investment. However, with a fast-changing environment, there has been an increasing number of investors who do not match with such traditional patterns. They are willing to invest and treat investments as a form of gambling to help them increase their earnings and incomes. Similarly, Waweru et al. (2008) concluded that the financial status of individuals generates influence on whether the investor invests their money, and also influences the amounts that they will be investing.

Risk-taking behaviour was found to have the ability to influence the investors' investment behaviours. This study also supports the past research findings that there is relationship between risk behaviour of investors and their investment behaviour and as such, the relationship changes for different individuals, such as individuals from higher socioeconomic classes or financial background are able to tolerate more risks due to their financial resources compared to investors from low socioeconomic classes that have limited financial resources in order to be able to efficiently and sufficiently invest in large sums of money in unit trust investments (Thaler 2015). The finding is also consistent with previous studies that the availability of the information is important for investors to influence their investment behaviours. This is crucial as accurate information will help investors with the necessary information that can be used by investors to make an informed investment decision.

However, this research has discovered that the investment revenue or expected return of investors has no relationship to the investors' investment behaviours and does not support findings in past research studies. Investors will consider the option that the past returns may not be significant to meet the current expected return in order to make investment decisions. In addition, current investors are more attracted by the investment options that deliver extraordinary returns to investors. Rapid economic growth has reliably increased the incomes and purchasing powers of individuals around the world and therefore, this has increased their desires and needs for wide varieties of products and services. At present, due to fast-changing environments, financial products have gained significant attention from individuals, and attracted individuals to make investment in different instruments in order to gain extra incomes and earnings, and investment has been used as an instrument by individuals as part of their personal financial planning.

This study has discovered that availability of information for investors has the strongest relationship to influence the investors' investment behaviours. The findings of this study were consistent with the previous studies. Availability of information influences investment behaviours as the investors are able to compile financial information to analyse and understand the specific instruments as well as the markets. The wide availability of information is perhaps the biggest benefit for the investors as they have an added advantage over other investors who may not have the complete information to make a comprehensive decision as well as to perform market analysis. Based on the findings, it is recommended that countries should advance the financial market by imposing stricter policies on the availability of information to investors. As such, the financial market needs to be regulated with stringent policies as well as the availability of providing information for investors to invest in unit trust funds. Fund managers have to provide sufficient information to the public in terms of both financial and non-financial information and update the information from time to time. This will help investors to make better investment decisions and will attract more investors to invest in unit trust funds.

On the other hand, investment brokers should develop a comprehensive clientele profile system that states the financial status of the investors, and the risk-taking behaviour of the investors. This system should be based on the analysis of the demographic profile of investors and nominate the number of investment options for investors' consideration and investment decisions. The utilization of technology could accurately perform analysis for investors and assist investors to make the informed investment decisions.

The first significance of this research finding is that it summarises the factors that are important to individual investors' investment behaviour for mutual fund investments. The finding could form the framework to guide new investors to invest in mutual fund, help them to choose, select, and consider the appropriate mutual fund for their investments to gain stable returns and incomes. In addition, the findings could also deliver valuable information to mutual fund brokers and managers, as they could use the information to attract individual investors to invest in mutual funds by providing the value for money types of mutual funds. This can be done by offering information about the benefits of mutual funds and the way in which these specific products can help investors meet their investment goals. These findings could also help government authorities to improve the development of the mutual fund sector by providing better investment environment for unit trust and help individual investors

to realize the benefits and merits of investments in mutual funds. This includes standardization of information in the mutual fund reports to ensure that more investors are confident to invest in the mutual funds rather than investing in risky stock markets. The study provides useful data to carry out researches on financial literacy among students which can be used to convince authorities to provide more financial courses in their university education programs. The findings can also be used by other and future researchers to develop their literature background and help them to understand the investors' investment behaviour in mutual funds. The findings of this research can also assist future researchers to compare the current findings and identify the changes that need to be incorporated in influencing investors' investment behaviour in mutual fund investments.

## 5. Conclusions and Recommendation

Rapid economic growth has reliably increased the incomes and purchasing powers of individuals around the world. Indeed, this has increased their desires and needs for wide variety of products and services. This research has been undertaken to discover the various behaviours shown by the investors in investing unit trust investments by analysing various factors that can greatly influence their investment decisions. In today's fast changing environments, financial products have gained significant attentions of individuals, and attracted many individuals to make investment in different instruments in order to gain extra incomes and earnings, and investments has been used as the instrument of individuals as part of their personal financial planning. Socio-economic factors are also significantly associated with investment behaviour. Therefore, unit trust management should take into consideration various socio-economic factors before introducing any new types of unit trust investments. This research investigates various factors that influence the investors' behaviours for unit trust investments. Investments in unit trusts funds are for a long-term period of time and most importantly it should be managed by professional fund managers that look for growth and performances of unit trust funds. Investors will also be encouraged to invest because of the accurate information provided to investors. Investors with sound financial status tend to invest in highly risky opportunities and exhibit more risk seeking behaviour as compared to less financial status investors. Risk-taking behaviour has also affected the investment decisions of the investors as the investor who takes high risks expects higher returns that are associated with their investments.

*Limitations and Future Research*

There are some limitations that are associated in this study, and these limitations may influence the generalization of the project findings, as well as influence the audience consideration on the usage of this research finding for their decision making. The sample size used in this study is relatively small in order to gain reliable insights of the research findings. On the other hand, the usage of the quantitative research method will not allow participants to express their opinions and thoughts freely. Future research should incorporate a qualitative research approach to allow investors to nominate factors that they consider as important to invest as well as to influence their investment behaviours and decisions, and this would allow the findings to be more useful and represent the behaviours of investors toward unit trusts. In addition, the sample size of the study needs to be increased as a larger sample group can yield more accurate study results. These research data were collected in Malaysia, and therefore the findings of this research are unable to represent investors of unit trust in other countries, as to the differences in culture and environment in other countries will cause different investment behaviours, and therefore future research requires the researcher to study in different cities or countries to ensure that it represents the general population. In addition, the findings of this research cannot give the full understanding of the investors' investment behaviours. The findings reveal that the factors involved in this research provide some influence on the investors' investment behaviour, and the audiences should not concentrate on these factors alone to make decisions, as there are other factors excluded that affect investment behaviours. Thus, future research needs to include more categories of factors for analysis such as investor's self-related factors, and external factors such as macro environment,

or factors relating to instruments, in order to provide a more comprehensive understanding of the investors' investment behaviour.

**Author Contributions:** S.A. and M.R. conceived of the presented idea. G.M. and A.K.L. developed the theory and performed the computations. S.A. and M.R. verified the analytical methods. G.M. and A.K.L. encouraged S.A. to investigate behaviour aspect specifically and supervised the findings of this work. All authors discussed the results and contributed to the final manuscript.

**Funding:** This research received no external funding.

**Conflicts of Interest:** The authors declare no conflict of interest.

## Appendix A Questionnaire

| | Statement | Strongly Disagree | Disagree | Neutral | Agree | Strongly Agree |
|---|---|---|---|---|---|---|
| **1. Financial Status** | | | | | | |
| 1.1 | My investment amounts based on my current financial status and incomes. | | | | | |
| 1.2 | I do not invest in instruments that are exceeds my current financial status. | | | | | |
| 1.3 | I make investment that is within my current affordability. | | | | | |
| 1.4 | I have monthly portion contributed from my incomes for the purpose of investment. | | | | | |
| 1.5 | The investment amounts varies according to my incomes. | | | | | |
| **2. Risk Behaviour** | | | | | | |
| 2.1 | I consider the risk of each type of instrument to make selection for the investment instrument. | | | | | |
| 2.2 | I invest in unit trust because it carries a lower level of risk compare with others. | | | | | |
| 2.3 | The level of risk determines the return from the investment. | | | | | |
| 2.4 | I took my personal risk assessment test to understand instruments that suits my risk-taking ability. | | | | | |
| 2.5 | The risk of the instrument determines my investment decisions. | | | | | |
| **3. Investment Revenue** | | | | | | |
| 3.1 | I have my own expected rate of return for the investment. | | | | | |
| 3.2 | I use my expected rate of return as the benchmark for choosing investment options. | | | | | |
| 3.3 | I will invest in a project that gives highest return. | | | | | |
| 3.4 | Unit trust has the nature to delivers lower investment revenue to investors. | | | | | |
| 3.5 | The investment revenue affects my investment decision for the instrument. | | | | | |

| | Statement | Strongly Disagree | Disagree | Neutral | Agree | Strongly Agree |
|---|---|---|---|---|---|---|
| **4. Sources of Investment Information** | | | | | | |
| 4.1 | I prefer to use information that are published by well-known organizations such as investment banks. | | | | | |
| 4.2 | I also use information that are published and analysed by others to assist my investment decision making. | | | | | |
| 4.3 | The past revenue and prices of the instrument is my prior focus for determining the potential of the instrument. | | | | | |
| 4.4 | The availability of information for a specific instrument affects my selection for the instrument. | | | | | |
| 4.5 | The source or channel of information for the instrument affects my investment decision | | | | | |
| **5. Investment Decision** | | | | | | |
| 5.1 | I will continue invest in unit trust in the future. | | | | | |
| 5.2 | I will introduce my friends and family members to take part for investing in unit trust. | | | | | |
| 5.3 | I invest in unit trust because it gives stable returns and revenues. | | | | | |
| 5.4 | Instead of depositing the money in banks, I will bear lower risks but earning higher returns than interests provided by banks. | | | | | |
| 5.5 | The unit trust is the part of my long term personal financial planning. | | | | | |

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
