# Peer review of "An Empirical Study on the Determinants of an Investor’s Decision in Unit Trust Investment"

_economies, doi:10.3390/economies7030080_

Round 1

Reviewer 1 Report

I thank the Editor for the opportunity to review “Factors Impacting an Investor’s Decision in Unit Trust Investment in Malaysia” submitted to the Economies.

The paper presents an interesting research topic. However, this work still needs of further major developments especially with reference to the abstract, introduction, methodological aspect and the conclusion section.

In particular, I suggest you to revise the following aspects:

I      suggest to standardize the abstract according to the standards of the Journal and to better define the objective in relation to the four research questions indicated. The purpose of this work feels again of a lack of clarity. It is not clear whether the authors wished to assess the impact or    identify the relationships between the various elements. Finally, it is not clear the originality of your work.

I      suggest to rewrite the introduction in order anticipate the methodology applied (mixed-methods), describing how your research is conducted, highlighting some preliminary results and literature gap. At the end, add  the remainder of your paper. In the actual version of the introduction  there are lots of sub-paragraphs that are not relevant and redundant.

The “methodology”  is not clear and I suggest to rewrite it. Is it a mixed method? I think so, but we need to indicate which type (e.g. Convergent parallel design, Explanatory sequential design, Exploratory sequential design,  Ebedded design, Multiphase design). I suggest to  read the following important papers: Creswell, J.W, Clark, V.L.P., Gutmann, M.L. and Hanson, W.E.      (2003), “Advanced mixed methods research design”, in Tashakkori A. and Teddlie, C. (Ed.), Handbook of mixed methods in social & behavioral research, Thousand Oaks, CA: Sage, pp. 209-240. Creswell, J.W. (1999), “Mixed method research: Introduction and application”, in Cijek T. (Ed.),  Handbook of educational policy, Academic Press, San Diego, CA, pp. 455-472. Ivankova, N.V., Creswell, J.W. and Stick, S.L. (2006), “Using Mixed-Methods Sequential Explanatory Design: From Theory to Practice”,  Field Methods, Vol. 18 No. 1, pp. 3-20. Finally, I think you should provide more information about the methodology of analysis that you have used, supported also by previously contributions in relation with the main aim of your research.

I      suggest to add the text of the survey questionnaire.

Some Authors indicated in the text (e.g. Creswell, 2009, Bodge, 1999) are not  into the references list and there are some misprints (uture, paragraph 7.2.)

The results are presented in a very dispersive and unclear way. I suggest inserting the related result for each research question.

Conclusion  section (paragraph 5, 6, 7) should be enriched in order to be useful for community of scholars.

Finally, the quality of communication is suitable but I suggest the Authors to make a proofreading.

Good luck for your work!

Author Response

Abstract have been reconstructed 

Gap has been identified

Methodology re written

Included  references in the text 

Analysis corrected

conclusion rewritten

Reviewer 2 Report

This study examines the determinants of Malaysian investors in investing in unit trusts. A survey was undertaken with a sample size of 202 respondents. The results suggest that investment revenue has little impact on investors’ investment decision making, whilst source of information does have some impact on investor’s investment decision making. This is an interesting study.

Major concerns

The motivation of this study is weak. The authors should further discuss the motivation of this study. Although the authors discuss the research problems. But I cannot see the reasons of doing this study as previous studies have done it before. Why we need this study? There are four research questions. But I believe these can be summarised into one overarching research question of what are the determinants of individual investors’ investment decision on unit trust investment. Nevertheless, the introduction failed to explain why we need to do this study. What is the gap in the literature? The gap should be the research question.

In addition, the literature review section can be enhanced.  It is also critical to determinate the contribution of this study. The significance of the study does not explain the contribution of this study to the literature. Although the findings could be useful for practitioners, what is the contribution of the study to the academic literature.

The literature review section and the introduction section have very similar information. Some sentences are identical. Why? This is too repetitive. I can’t see the rationale of doing it.

The regression results do not make sense to me. For instance, the t- values of risk behaviour and financial status are 0.791 and -0.939 respective. But the p-values for both variables are 0.03 and 0.049 respectively. These do not make sense. The low t-values SHOULD NOT be significant. Please check your results. These again lead to a misleading discussion on the results, as well as the conclusion of the study.

The methodologies can be improved. Please control other variables such as education levels etc.

Minor comments:

1)      The article is too long and too repetitive.

2)      The article should be considerably condensed, yet an enhanced for the writing is required.

3)      Proof reading is required as main editorial mistakes such as section 7.2 “future” and ‘f’ is missing. Abstract, the first sentence: Unit trust is one of the major type of financial instrument…. Check the sentence.

4)      Page 11 suggested that respondents aged between 25 to 60 years old. BUT page 17 shows there are two respondents aged more than 61 years old. Which part is correct?

5)      Discuss the respondent rate.

Author Response

Repetitive has been deleted

Rewritten comprehensively

Proof reading done

incorporated response rate

analysis has been rewritten 

Round 2

Reviewer 1 Report

I thank the Editor for the opportunity to review the new version of the paper “An analysis on the factors Influencing Investor’s Decision in Unit Trust Investment in Malaysia” submitted to the Economies. I appreciated the authors' effort in writing important parts of the paper (abstract and discussion/findings). However, the work still needs an effort with reference to methodology and the literature review.

In particular,

The methodology should be supported by previous studies on the topic. For example,      “This research uses the non-probability sampling method, which is the convenient sampling technique used to allow researcher to select and choose participants at the best convenience of the researcher. This helps to reduce the time spent for selection of participants”, please add some reference.

The paragraph on the literature review is poor and the phrase “Obamuyi (2013) study indicated that socio-economic factors influence investment decisions of 123 investors in the capital market” it is not connected with the next part.The literature review should present a focused and a carefully structured outline of what others (academics/researchers) have      done in your topic area.

As previously written I suggest to add the text of the survey questionnaire.

Best regards

Author Response

Proof reading undertaken to solve all issues on language

Research design justified

Conclusion re-written 

All issues highlighted have been taken into consideration and solved accordingly 

Title has been changed to reflect the manuscript content 

An empirical study on the determinants of investor’s decision in unit trust investment
